

# A theory of abrupt climate changes: their genesis and anatomy
**Hsien-Wang Ou**[1]
[1] Lamont-Doherty Earth Observatory, Columbia University, Palisades, NY10964, USA
Corresponding author: Hsien-Wang Ou (hsienou0905@gmail.com)
**Abstract.** We integrate our previous ice-sheet and climate models to examine abrupt climate
changes pertaining to Heinrich event (HE), Dansgaard-Oeschger (DO) cycle as well as last de-
glaciation punctuated by Younger Dryas (YD). Since they are all accompanied by ice-rafted de-
bris, we posit their common origin in the calving of ice sheet due to thermal switch at its bed.
Such thermal switch would generate step-like freshwater flux and together with decadal ocean
response, they would endow abruptness to these millennial climate signals, which need not in-
volve ocean mode change, as commonly assumed. We distinguish thermal switches due to geo-
thermal heat and surface melt, which would calve inland/marginal ice to drive HE/DO-cycle, re-
spectively. As such, the glacial DO-cycle hinges on post-HE warmth that enables the ablation
whereas the Holocene DO-cycle is self-sustaining. The ocean response to freshwater flux entails
millennial adjustment to maximum entropy production (MEP), a process termed "MEP adjust-
ment". As its direct consequence, the termination of HE is accompanied by sudden warming fol-
lowed by gradual cooling to exhibit saw-toothed H-cycle, and the cooling moreover would an-
chor DO-cycles to form the hierarchical Bond cycle. The meltwater produced during deglacia-
tion, if rerouted to Hudson Bay, may augment the calving-induced freshwater flux to cause YD,
the latter thus involves happenstance and did not materialize during penultimate deglaciation.
By incorporating calving origin of the freshwater flux and MEP adjustment of the ocean, the the-
ory has provided an integral account of these abrupt climate changes.
**Non-technical summary**
We integrate our previous models to examine abrupt climate changes pertaining to Heinrich and
Dansgaard-Oeschger cycles as well as deglaciation punctuated by Younger Dryas. We posit
their common origin in the calving of ice sheet triggered by thermal switches at its bed, which
are differentiated between that caused by geothermal heat and surface melt. Together with the
ocean evolution toward maximum entropy production, the theory provides an integrated account
of observed phenomena.
**1. Introduction**
Last ice age was teemed with abrupt climate changes pertaining to Heinrich (H) events (HE),
Dansgaard-Oeschger (DO) cycles as well as deglaciation punctuated by Younger Dryas (YD), a
dramatic climate reversal. While these climate signals are distinct, they are all accompanied by
ice-rafted debris (IRD, Bond et al. 1997, Fig. 6), suggesting a common origin in the calving of
ice sheet due to thermal switch at its bed (MacAyeal 1993; Ou 2022a). Since recurring time of
calving is constrained by ice mass balance, the resulting freshwater flux is naturally availed the
millennial timescale, a timescale not inherent to the ocean. On the other hand, thermal switch is





operating on very short (years) subglacial hydrological timescale (Fricker et al. 2007) to render
step-like freshwater flux, and together with the decadal ocean response (Duplessy et al. 1991),
they endow abruptness to the millennial climate signal, which thus need not involve mode
change, as commonly assumed (Alley et al. 2003). Besides the abruptness, the presumed mode
change is also prompted by large signal in the surface-air temperature (SAT), which however
may simply reflect extremely cold winter air during stadials characterized by extensive sea-ice
cover (Denton et al. 2005) whereas variation in the sea-surface temperature (SST) remains well
short of mode change except during deglaciation (Bard 2002).
With above common origin in the thermal switch at the ice bed, we nonetheless distinguish their
two differing heat sources hence locales: for H-cycle, it is the trapping of the geothermal heat by
the growing ice sheet that leads to calving of the inland ice (MacAyeal 1993); and for DO-cycle,
it is the surface melt over the ablation zone (Hooke 1977) that causes calving of the marginal ice.
Both these thermal switches hence differing calving processes have been demonstrated by nu-
merical calculations (Calov et al. 2002; Brinkerhoff and Johnson 2015), and our ice-sheet model
(Ou 2022a) allows a prognosis of the resulting freshwater flux, which thus may be prescribed as
external perturbation of the ocean.
To examine the ocean response, we shall apply our box model of coupled ocean/atmosphere (Ou
2018). As deduced therein, since the meridional overturning circulation (MOC) involves random
eddy exchange across the subtropical front, a generalized second law of nonequilibrium thermo-
dynamics (NT) leads to its millennial evolution toward maximum entropy production (MEP), a
process termed "MEP adjustment". Incorporating this process, Ou (2022b, under review) has ex-
amined the ocean response to the orbital forcing and shows that it can reproduce Pleistocene gla-
cial cycles while resolving many longstanding puzzles, in support of its utility. In the present
theory, we shall show that combining calving of the ice sheet and MEP adjustment of the ocean,
it may provide an integrated account of abrupt climate changes.
For self-containment, we first discuss in Sects. 2 and 3 the relevant physics of our ice-sheet and
climate models, respectively, which is then applied in Sects. 4 to 6 to examine successively ab-
rupt climate changes pertaining to H- and DO-cycle as well as deglaciation. In each, we high-
light salient features of the observed phenomena, discuss their genesis based on model physics,
and provide a synthesis of previous works. We summarize main findings of the theory in Sect. 7
to conclude the paper.
**2. Ice-sheet model**
We first discuss our ice-sheet model (Ou 2022a), which provides the freshwater perturbation to
the ocean. It is well recognized that large ice sheet may exhibit quasi-periodical surge due to
thermal switch at its bed (MacAyeal 1993). Physically, ice growth by accumulation would in-
creasingly trap the geothermal heat to warm the bed to the pressure-melting point when a surge is
triggered; the ensuing thinning would augment the conductive cooling to refreeze the bed, termi-
nating the surge. As the thermal switch is also favored by greater driving stress, it would be sited





off the ice divide to calve inland ice through Hudson Strait, which has been widely attributed as
the source of H-events (Bond et al. 1992). The numerical simulation of the ice discharge how-
ever often involves tuning of the sliding velocity (Calov et al. 2002), which directly impacts the
amplitude and period of the surge cycle. To remove this empiricism, we have incorporated
global momentum balance to constrain the sliding velocity by the strait width (Tulaczyk et al.
2000), so the model closure allows us to prognose the surge properties.
A tangible outcome of the model is the construction of a 2-D regime diagram as sketched in Fig.
1, which is spanned by scaled length ($l$) and width ($w$) of the ice discharge and on which surge
properties, such as termination height ($h$), surge/creep duration ($t_s/t_c$), and surge velocity ($u$),
can be contoured (all nondimensionalized). It is seen that the model has delineated three dynam-
ical regimes: steady creep, steady sliding and cyclic surge separated by thick and shaded lines,
which can be understood as follows. For short discharge, the catchment due to accumulation can
be absorbed by creep to maintain a steady state, so the thermal switch remains off. For longer
discharge, the thermal switch would turn on to trigger the sliding motion whose strength how-
ever depends on the strait width: for a narrower strait hence slower sliding, the ice flux can be
sustained by catchment to maintain a steady state, but for a wider strait hence faster sliding, the
ice flux cannot be sustained, thus vaulting into surge cycle. The box marked H represents ice
discharge through the Hudson Strait, which falls well within the surge regime, and the prognosed
surge properties are of the same order as that inferred from observation, in support of ice-calving
source of the H-cycle.

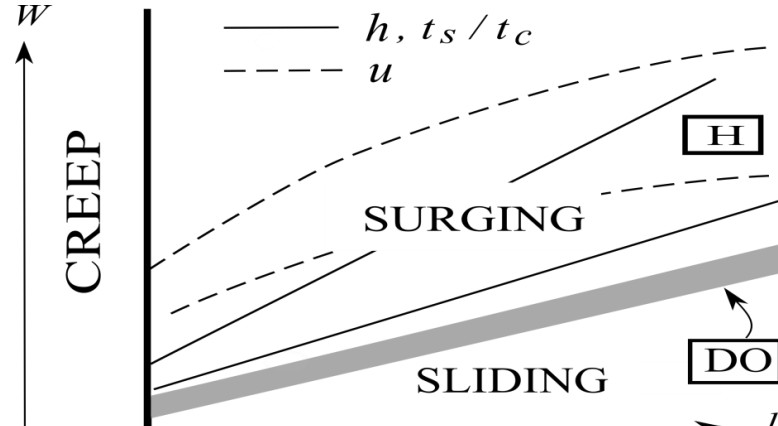


Fig. 1. A schematic of the regime diagram spanned by the scaled length $l$ and width $w$ of
the ice discharge, which consists of steady-creep, steady-sliding and cyclic-surge regimes
separated by thick and shaded lines. Contoured surge properties (thin lines) include ter-
mination height ($h$), surge/creep duration ($t_s/t_c$), and surge velocity ($u$). Box H marks
the ice discharge through Hudson Strait, and the shaded line indicates its aspect ratio for
the DO-cycle.



Since DO-cycle is associated with much smaller freshwater flux than H-cycle, it likely involves
calving of the marginal ice. Searching for clues that might differentiate the two, we note that en-
glacial ice-sheet temperature shows two distinct zones of temperate bed (Hooke 1977, Fig. 4d):
besides the one under ice divide due to trapping of the geothermal heat, there is another one in
the ablation zone where surface melt is particularly effective in warming the bed via vertical ad-
vection. We posit therefore that DO-cycle is driven by the thermal switch under the ablation
zone, which would calve the marginal ice. Unlike ice discharge through Hudson Strait, this calv-
ing may occur along the eastern seaboard of Laurentide ice sheet (LIS) hence unconstrained by
topography, and numerical calculations of ice discharge over a flatbed (Brinkerhoff and Johnson
2015) provides an apt demonstration of its plausible scenario: following the thermal trigger, the
surging ice would grow in width until it is arrested by a limit cycle, resulting in periodic self-or-
ganized ice streams.
Since the thermal switch underlying Fig. 1 is generic, the stream width arrested by limit cycle is
precisely that divides the steady-sliding and cyclic-surge regimes (shaded). Denoting the corre-
sponding stream properties by subscript "s", they are functions only of the "heating" parameter
measuring the relative strength of the frictional to geothermal heating
$$\alpha_h = \rho_i g \dot{a}[h]/\dot{g} \tag{1}$$

where $\rho_i$ is the ice density, $g$, the gravitational acceleration, $\dot{a}$, the accumulation, $[h]$, the ice
thickness scale taken to be equilibrium-line altitude (ELA), and $\dot{g}$, the geothermal flux. Specifi-
cally, the termination height is
$$h_s = (\sqrt{1 + 2\alpha_h} - 1)/\alpha_h, \tag{2}$$

the aspect ratio is
$$a_s = \sqrt{2}/h_s, \tag{3}$$

and the ratio of surge/creep durations is
$$t_{ratio} = 2(1 - a_s^2)^{-1} \tag{4}$$

for which we have set the mean thinning rate of the surge to be half its maximum. They are plot-
ted in Fig. 2 whose qualitative dependence can be explained as follows: for stronger frictional
heating, the ice would be thinner before the conductive cooling may terminate the surge, which
in turn would be wider on account of the mass balance, and then such wider surge implies faster
sliding motion to shorten the surge relative to the creep phases. Applying standard values listed
in Appendix, the heating parameter is .48 (dashed), which yields a fractional surface depression
of .17, quite smaller than that of HE (about .5, see Ou 2022a). The surge and creep have compa-





rable duration ($t_{ratio} = 1.1$), both lasting about 1 ky. In comparison with the saw-toothed H-cy-
cle, the shorter creep is due to the smaller surface depression during surge, which needs less time
to be replenished by accumulation, but the surge duration is maintained by the slower discharge
hence thinning.

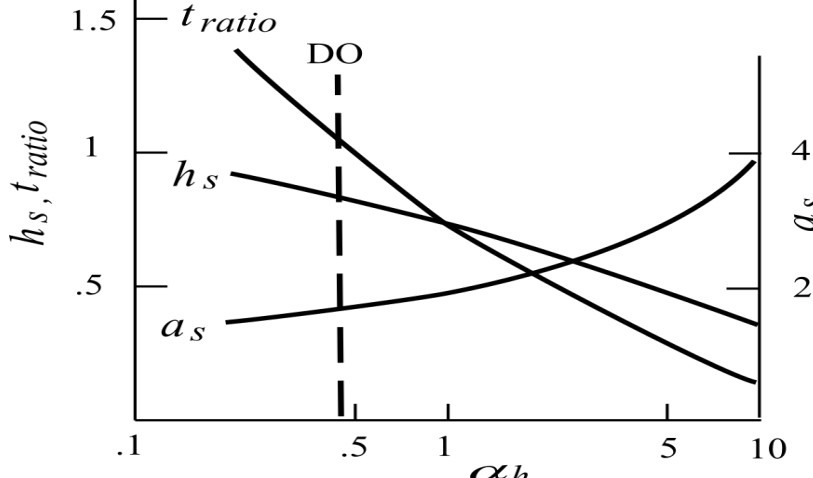


**Figure 2**: Ice-discharge properties of the DO-cycle plotted against the heating parameter.
They are the termination height $h_s$, aspect ratio $a_s$ and ratio of surge/creep durations
$t_{ratio}$, all nondimensionalized. The vertical dashed line is representative of the DO-
cycle, which shows comparable surge/creep durations.
To recap, our ice-sheet model has produced surge properties required by H- and DO-cycles, so
the resulting freshwater flux may be prescribed as external to the ocean to examine its response,
as discussed next.
**3. Climate model**
Readers are referred to Ou (2018) for detailed derivation of our coupled climate model, only rel-
evant physics is summarized here for self-containment. The model configuration is sketched in
Fig. 3 for which both planetary fluids are composed of warm/cold boxes aligned at mid-latitudes
and the adjacent ice sheet injects freshwater perturbation into the subpolar water, as determined
from our ice-sheet model. Retaining dominant balances, the absorbed SW flux ($q$) differentiates
the SST ($T$), which differentiates the SAT by the convective flux ($q_c$) to induce atmospheric heat
transport; the attendant moisture transport depresses the subpolar salinity (S), which together
with the SST specifies the density surplus ($\rho$); the latter drives the MOC ($K$) across the subtropi-
cal front composed partly of random eddy shedding. Since climate signals of our concern are



dominated by that of the cold-box, the model variables are the nondimensionalized cold-box de-
viations from (known) global means.

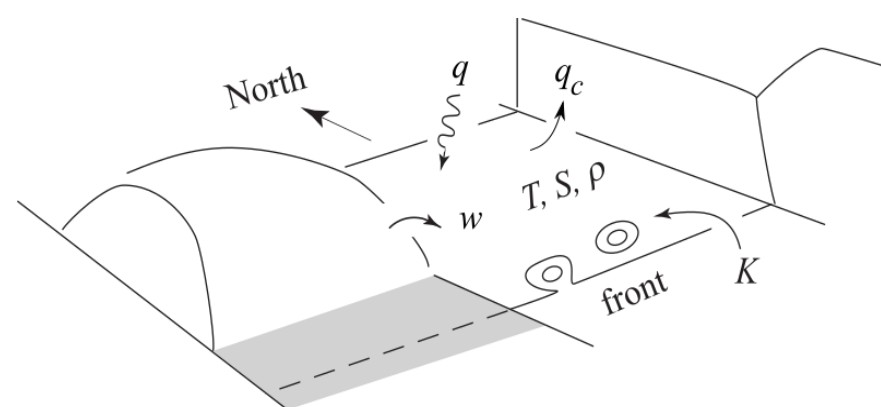


**Figure 3:** The model configuration of coupled ocean/atmosphere composed of warm and
cold boxes aligned at mi-latitudes and ice sheet on the adjacent continent providing fresh-
water perturbation to the subpolar ocean. Model variables are the cold-box deviations
from (known) global means and the MOC.
The model physics can be illustrated via a phase-space diagram sketched in Fig. 4 whereby the
cold-box temperature deficit ($T$) and density surplus ($\rho$) are plotted against MOC ($K$). Expect-
edly, decreasing MOC cools the subpolar water (that is, increasing temperature deficit) and re-
duces the convective flux from its global mean ($\bar{q}_c$), but since the convective flux may not be
negative, the ocean/atmosphere coupling renders a "convective bound" at
$$T = T_c \equiv 2\bar{q}_c ,\qquad\qquad\qquad (5)$$
which divides the climate regime into warm/cold branches with a break in the density slope.
Since the cold branch is characterized by vanishing convective flux, the atmospheric heat
transport is saturated at $\bar{q}_c$. This division into warm/cold branches is an outcome of atmospheric
coupling, which unlike ocean-only models (Stommel 1961) allows normal-signed density con-
trast, as seen in coupled climate models (Rahmstorf et al. 2005). That the cold branch is charac-
terized by vanishing convective flux is consistent with diagnosis of the cold state from such
models (Manabe and Stouffer 1988, their Fig. 18), in support of the convective bound.





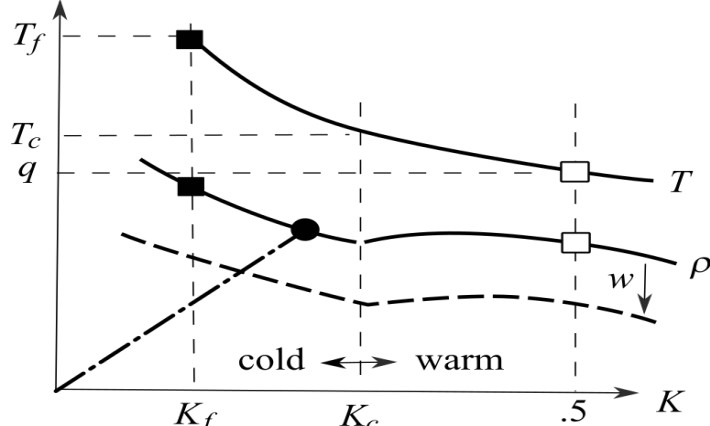


**Figure 4:** A schematic of the regime diagram whereby subpolar temperature deficit ($T$)
and density surplus ($\rho$) are plotted against MOC ($K$). The convective bound (subscripted
$c$) separates warm/cold branches with their respective MEP (rectangles). The glacial
MEP (solid rectangles) is defined by the freezing-point (subscripted $f$) subpolar tempera-
ture. The MOC line (dash-dotted) pivots on millennial timescale toward MEP, whose
intersect with the density curve specifies the ocean state (oval). The density curve is dis-
placed (thick-dashed) instantaneously by the freshwater flux ($w$).

To constrain MOC, we assume it to be proportional to the density surplus (Stommel 1961), as
indicated by dash-dotted line (referred as "MOC line") whose intersect with the density curve
then specifies the climate state (oval). In numerical models that do not resolve eddies, the MOC
line is fixed by diapycnal diffusivity, which in effect is a free parameter finely tuned to yield the
observed state (Rahmstorf et al. 2005), but the actual MOC is subjected to random eddy ex-
change across the subtropical front (Auer 1987; Lozier 2010) and applying the fluctuation theo-
rem --- a generalized second law (Crooks 1999), we deduce that the MOC line would pivot on
millennial timescale toward MEP, a process termed "MEP adjustment". There can be MEP in
both warm/cold branches, which are referred as interglacial (open rectangles) and glacial (solid
rectangles) MEPs, respectively.
The interglacial MEP is derived to be $(T, K = q, 1/2)$, which is consistent with the observed in-
terglacial; and the glacial MEP is specified by freezing-point subpolar water $(T = T_f)$ that is free
of perennial ice. The reason for the latter is because such ice would curb the ocean cooling to
weaken the MOC --- in contradiction to MEP. This glacial MEP is consistent with that observed
during last glacial maximum (LGM) characterized by freezing-point subpolar water (Kucera et
al. 2005), which remains open in summer (de Vernal et al. 2005). Since MEP adjustment oper-
ates on millennial timescale, the above deduction does not preclude perennial sea ice during ab-
rupt climate change, and then with SST being at the freezing point, extensive sea ice necessarily
forms in winter, both as seen later.





The freshwater flux from the calving ice would depress the density curve as indicated by an ar-
row toward the dashed line. The displacement of the density curve is assumed instantaneous rel-
ative to the millennial climate signals (Sect. 1). On the other hand, the orbital forcing hence the
temperature curve may be taken as unvarying through abrupt climate changes.
**4. H-cycle**
**4.1 Phenomenology**
The last glacial was punctuated by recurring Heinrich events (HE) when massive calving of ice-
bergs strewed IRD across the North Atlantic (Heinrich 1988; Bond et al. 1992). As discussed in
Sect. 1, the onset and termination of HEs are abrupt relative to their millennial duration spaced
several millennia apart (Elliot et al. 2002). The accompanying freshwater flux is substantial,
amounting to sea-level change of O (10 m) (Chappell 2002), which further depresses MOC from
its already weak glacial strength (Elliot et al. 2002). Since the subpolar water is already at the
freezing point (Kucera et al. 2005), a weakening of MOC causes formation of extensive sea ice
to maintain the ocean heat balance (Broecker 1994). The ice-covered ocean deters melting of
icebergs, allowing IRD to spread to the glacial polar front (Grousset et al. 1993; van Kreveld et
al. 2000).
The MOC resumes at the termination of HE, the subpolar water however does not merely return
to the pre-HE state of freezing-point temperature, but a state several degrees warmer (Bard
2002). This post-HE warming is followed by gradual cooling to the pre-HE glacial state, thus
forming the saw-toothed H-cycle (Alley 1998; Henry et al. 2016) --- albeit the cooling trend is
populated by millennial DO-cycles. While SAT signal associated with H-cycle spans O (10 $^0$C),
the SST variation is considerably smaller, ranging in low single digits (Alley 1998; Bard 2002).
The substantial change in MOC has led to anti-phased Antarctic climate during HE, outside of
which the hemispheric climates remain synchronous (Broecker 1998; Clark et al. 1999).
**4.2 Genesis**
As discussed in our ice-sheet model, we posit HE to originate from calving of the inland ice
through Hudson Strait. We illustrate the ocean response to the freshwater flux via a phase-space
diagram in Fig. 5 with the resulting timeseries shown in Fig. 6. As a representative example, we
set the annual absorbed SW flux at 90 $W\ m^{-2}$ below the global mean, global convective flux at
56 $W\ m^{-2}$, global-mean SST at 14 $^0$C and anomalous freshwater flux at .1 $Sv$, so for scales de-
fined in Appendix, they yield dimensionless parameters $\left(q, \bar{q}_c, T_f, w\right) = (0.9, 0.56, 1.75, 0.05)$;
and for the time series, we set the surge/creep duration at 1/5 ky with light/dark shades symboliz-
ing freshwater flux and sea-ice cover, respectively.



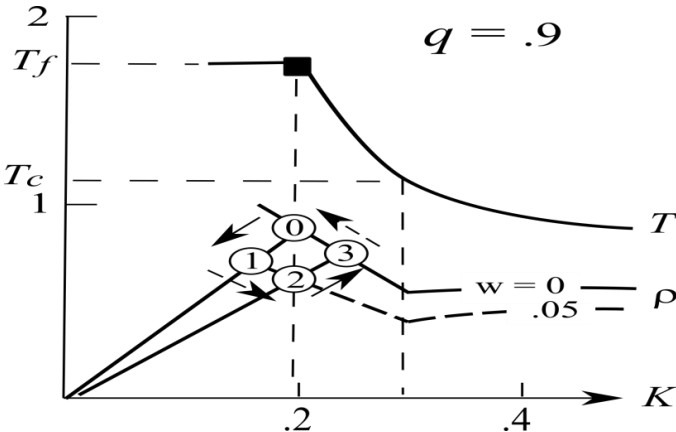

**Figure 5:** The H-cycle in the phase-space diagram when the freshwater flux displaces the density curve (thick dashed). The cycle goes through numbered states with solid arrows indicating abrupt changes and dashed arrows, the millennial adjustment to MEP (solid rectangle).

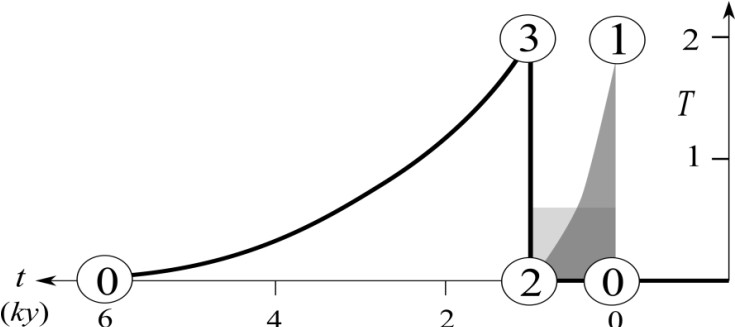

**Figure 6:** A schematic of H-cycle in the time domain corresponding to that of Fig. 5. Light and dark shades symbolize freshwater flux and sea-ice extent, respectively.

The H-cycle begins with the glacial MEP (State 0) when the thermal switch is turned on and runs through numbered states with solid and dashed arrows indicating fast (decadal) and slow (millennial) transits, respectively. Since the MOC line pivots on millennial timescale, it remains immobile at the HE onset, so State 0 would transition to State 1 whereby the weaker MOC would induce extensive sea ice (dark-shaded in the time plot) to maintain the ocean heat balance. During the millennial HE, the MOC line would pivot toward MEP by melting the sea ice (Sect. 3) hence transition State 1 to 2. At the HE termination, the MOC line is again immobile to transition State 2 to 3, which thus exhibits a sudden warming. During the ensuing creep phase, the MEP adjustment would pivot the MOC line toward the glacial MEP (State 0), thus evincing a gradual cooling to form the saw-toothed H-cycle.



A key property of the H-cycle is the post-HE warmth, which can be derived as follows. As the
H-cycle resides in the cold branch for which the atmospheric heat transport has saturated at the
global convective flux $\bar{q}_c$ (Sect. 3), the total heat transport (given by the orbital forcing $q$) is par-
titioned between atmosphere and ocean as
$$q = \bar{q}_c + KT \,, \tag{6}$$

and the salinity balance states
$$\mu \bar{q}_c + w = KS \tag{7}$$

where $\mu$ is a moisture parameter so the first term is the atmospheric moisture transport, which to-
gether with the freshwater flux $w$ is balanced by the salinity flux carried by MOC. Combining
these two equations yields the density surplus
$$\rho = T - S \tag{8}$$

$$= \frac{1}{K}(q_e - w) \,, \tag{9}$$

where
$$q_e = q - (1 + \mu)\bar{q}_c \tag{10}$$

is the forcing in excess of the warm-transition threshold (the last term), the latter being the maxi-
mum insolation (that is, $w = 0$) when the ocean may stay in the cold branch (that is, $\rho \geq 0$).
From Eqs. (6), (9) and trigonometry, we derive that
$$\frac{T_3}{T_2} = \frac{K_2}{K_3} \tag{11}$$

$$= \frac{\rho_2}{\rho_3} \tag{12}$$

$$= \frac{K_3}{K_2} \cdot \frac{q_e - w}{q_e} \,, \tag{13}$$

so Eqs. (11) and (13) yield
$$\frac{K_2}{K_3} = (1 - \frac{w}{q_e})^{1/2} \,. \tag{14}$$

Substituting Eqs. (14) into (11), we arrive at



$$\frac{T_3}{T_2} = (1 - \frac{w}{q_e})^{1/2} \qquad (15)$$
The (dimensional) temperature range ($\Delta T$) of the H-cycle thus is
$$\Delta T = [T](T_2 - T_3)$$
$$\approx \bar{T} \cdot \frac{w}{2 \, q_e} , \qquad (16)$$
for which we have assumed $w/q_e \ll 1$. For parameter values specified earlier, $q_e = .17$, so
$w/q_e = .29$, the approximation Eq. (16) thus yields $\Delta T \approx 2 \, ^0$C (as shown in Fig. 6), which repre-
sents an underestimate of about 10 %. Given crudeness of the model, we prognose therefore a
post-HE warmth in low single digits, which is commensurate with the observed one (Bard 2002).
Since this warmth increases with the freshwater flux and summer insolation, it would lead to de-
glaciation when certain threshold is exceeded, a topic to be discussed in Sect. 6.
**4.3 Synthesis**
The post-HE warming has been attributed to resumption of MOC, which however does not
merely return the subpolar water to the pre-HE state but a few degrees warmer, and the subse-
quent cooling has been ascribed to the downwind effect of growing LIS during the binge phase
(Alley 1998), whose efficacy remains to be demonstrated (Clark 1992). In our interpretation, on
the other hand, both these features are direct consequences of the MEP adjustment: by melting
sea ice during HE, it flattens the MOC line, which necessarily yields a warmer state at the termi-
nation of HE; then the same process would cool the subpolar water toward the glacial MEP de-
fined by freezing point; there is no need to invoke disparate or extraneous physics.
With the gradual cooling being the climate response to HE, it does not precondition HE (Alley
and Clark 1999), which runs on the internal ice clock (Sect. 2). On the other hand, the climate
response to the primary ice calving through Hudson Strait would synchronize calving of lesser
ice sheets around the North Atlantic to augment the freshwater flux (Grousset et al. 1993). The
sea-ice cover during HE is induced by MOC weakened by freshening, so it is a continuous func-
tion of the freshwater flux, which moreover is dissipating through HE; a distinct H-mode from
the glacial state thus cannot be defined nor is it necessary (Alley and Clark 1999).
Because of abruptness of the post-H warming and the large annual SAT signal registered in the
ice core, H-cycle has been modelled as ocean mode change (Paillard 1995; Ganopolski and
Rahmstorf 2001), which is unsupported by observation: the abruptness merely reflects the deca-
dal ocean response to step-like freshwater perturbation (Sect. 1) and the large SAT signal stems
from the extremely cold winter air during HE due to extensive sea ice cover (Broecker 1994; Li
et al. 2010). In contrast to SAT, the SST signal ranges only in low single digits (Bard 2002) and
is proportional to the freshwater perturbation Eq. (16), both negating their interpretation as mode



change.  While short of mode change, the abrupt MOC change associated with HE is nonetheless
of sufficient magnitude (Elliot et al. 2002) to induce anti-phased Antarctic climate (Broecker
1998; Clark et al. 1999).  Outside HE however, hemispheric climates remain synchronized by
global teleconnection (Broecker 1998).
**5.  DO-cycle**
**5.1 Phenomenology**
The cooling phase of H-cycle is populated by millennial-scale DO-cycles to form bundled Bond
cycle (Dansgaard et al. 1993; Bond et al. 1993).  Its hierarchical structure is intriguing: DO-
cycles emerge only after post-HE warming, and then both their stades/interstades follow the H-
cooling trend.  The stades are accompanied by IRD (Bond et al. 1997; van Kreveld et al. 2000),
suggesting their origin in ice calving, which endows abruptness to the millennial-scale DO-cycle.
Despite this common origin in ice calving, DO-cycle differs from H-cycle in important respects:
first, the freshwater flux is considerably smaller (Yokoyama and Esat 2011), suggesting calving
only of the marginal ice (Sect. 2); second, while SAT range remains large, SST and MOC signals
are further muted and no hemispheric linkage can be discerned (Bond et al. 1995; Charles et al.
1996; Elliot et al. 2002); third, unlike saw-toothed H-cycle, DO-cycle is more symmetric with
comparable millennial duration for both stades and interstades (Alley 1998).
Perhaps the most significant observation of DO-cycle is its prevalence in Holocene even in the
absence of large ice sheet and H-cycle (Bond et al. 1997).  Yet, the Holocene DO-cycle is still
accompanied by IRD and retains similar time signature as its glacial counterpart, suggesting their
common genesis (Bond et al. 1997).  Without sea ice covering the subpolar ocean, the SAT
range of Holocene DO-cycle is much reduced, and SST and MOC signals are indistinct (Grootes
and Stuiver 1997), which certainly are unrelated to mode change; the common genesis of the
Holocene and glacial DO-cycle thus further degrades the latter's interpretation as involving
mode change.
**5.2 Genesis**
We have posited in Sect. 2 that DO-cycle has its origin in the periodic calving of the marginal ice
of the ablation zone.  Since during glacial time, there can be ablation only by the post-H warmth,
the glacial DO-cycle thus is preconditioned on H-events and anchored on the cooling phase of
the H-cycle.  The genesis of the DO-cycle is illustrated in the phase space in Fig. 7, which thus is
enclosed within the H-cycle marked by the thin outer line.  Following the same convention as the
H-cycle, the DO-cycle goes through numbered states with solid and dashed arrows indicating
fast (decadal) ocean response and slow (millennial) MEP adjustment, respectively, and its time
series is plotted in Fig. 8, for which we have assumed square-wave freshwater flux of 2 ky period
(shaded columns) and dark shades symbolize the sea-ice cover.




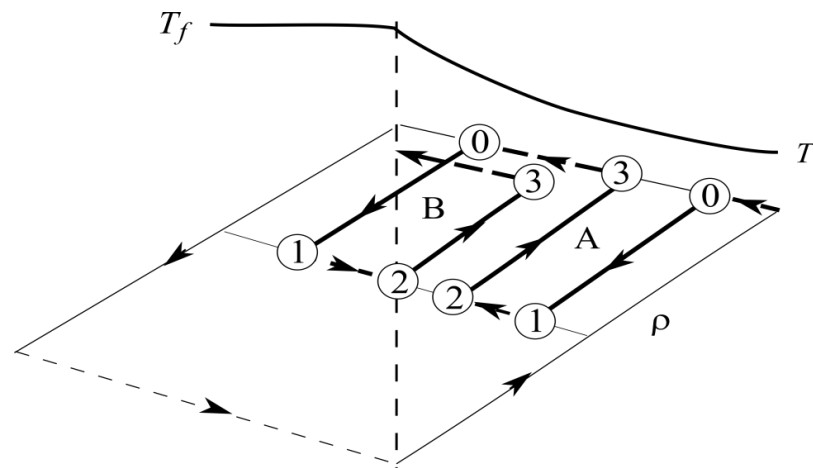

**Figure 7:** Same as Fig. 5, but for DO-cycles anchored on the cooling phase of H-cycle
(thin lines). Type-A's stade remains above the freezing point while type-B's has reached
the freezing point to resemble a mini-H-cycle.


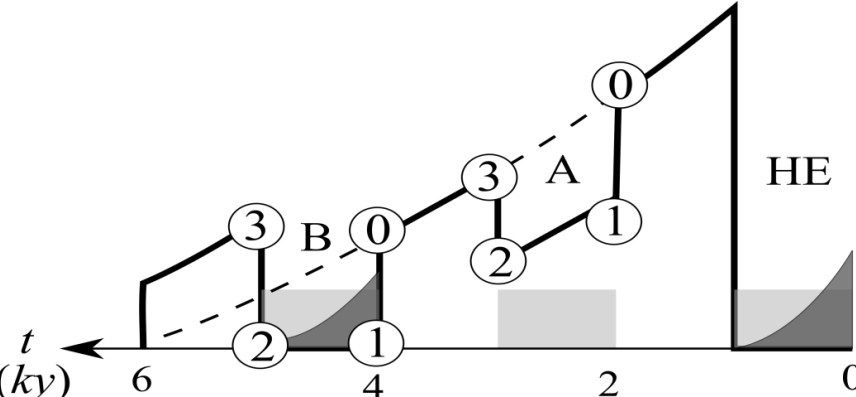


**Figure 8:** A schematic of a preceding HE and DO-cycles in the time domain correspond-
ing to that of Fig. 7. Light and dark shades indicate freshwater fluxes (a square-wave of
2 ky period) and sea-ice cover. Type B's stade has reached freezing-point to cause for-
mation of the sea-ice and a greater rebound of the ensuing interstade, but otherwise DO-
cycles trend as the H-cooling curve to exhibit the hierarchical Bond cycle.





Unlike H-cycle, we need to distinguish two types of DO cycle, designated as type-A and B in the
figures.  For type-A, its stade remains above the freezing point hence unobstructed in its trending
of the H-cooling, so the ensuing interstade simply returns to the H-cooling curve in the time plot.
For type-B however, its stade has reached the freezing point to cause formation of sea ice, just
like the H-cycle, so the ensuing interstade would protrude above the cooling curve, resembling a
miniature post-H warming.  Despite the protrusion, both stades/interstades trend as H-cooling to
exhibit hierarchical structure of the Bond cycle, as depicted in Alley (1998, Fig. 1).
While timing signature of the DO-cycle is controlled by internal ice dynamics, its initial trigger
is due to the post-HE warmth whose timing thus is related to vertical advection associated with
surface melt.  As a cursory estimate, a summer melt rate of 2 $m\ y^{-1}$ (Oerlemans 1991) would
yield vertical- and annual-averaged vertical velocity of O (0.5 $m\ y^{-1}$) to render an advective
timescale of O (1 ky) hence it need not be differentiated from the creep duration.

**5.3 Synthesis**

Since DO-cycle is accompanied by IRD (Bond et al. 1997), we posit that it is originated in ice
calving, just like HE except the thermal switch lies under the ablation zone to calve the marginal
ice.  This origin avails the DO-cycle with step-like freshwater flux of millennial duration --- its
defining characteristics in common with H-cycle; but differing from the latter, the glacial DO-
cycle commences only after the post-HE warmth has set up the ablation zone to activate its ther-
mal switch, the reason that the glacial DO-cycle is encased within H-cycle to form the hierar-
chical Bond cycle.  In the interglacial, on the other hand, the ablation zone is already in existence
around Greenland ice sheet (Oerlemans 1991), so DO-cycle would be self-sustaining and retain
the same time signature as its glacial counterpart.  This commonality thus stems from ice dynam-
ics of the ablation zone, not the large ice sheet whose absence in Holocene has led Bond et al.
(1997) to conjecture (unknown) climate origin of the DO-cycle.
The large SAT signal of the glacial DO-cycle has been widely interpreted as indicative of ocean
mode change (Broecker et al. 1990), which however may simply reflect the vast sea-ice cover
during DO-stades that strongly cools the winter air (Denton et al. 2005; Li et al. 2010) whereas
the SST signal remains short of mode change.  Since there is little sea ice in the interglacial, the
SAT signal of the Holocene DO-cycle is much reduced, and the SST variation is further muted
and obviously is unrelated to mode change; its common genesis with the glacial DO-cycle would
further militate against the latter's interpretation as involving mode change.
Besides the mode change (Ganopolski and Rahmstorf 2001) critiqued above, the DO-cycle has
also been modelled as damped oscillation when the ocean is perturbed by freshwater forcing or
noise (Sakai and Peltier 1999; Schulz and Paul 2002).  The modelled period however depends
critically on the strength of the perturbation, which is difficult to reconcile with the comparable
time signature of Holocene and glacial DO-cycles: not only is there no anomalous freshwater
flux in the interglacial, the HE-induced freshwater flux has already ceased before the onset of
glacial DO-cycle for which the freshwater flux is its manifested cyclic signal, not an external





stimulus. In our interpretation, the timing signature of DO-cycle is deterministic and set by the
ice dynamics of the ablation zone, and the ocean role is relegated to setting-up the ablation zone
in enabling the DO-cycle.
**6.  Deglaciation**
**6.1 Phenomenology**
The most dramatic abrupt change occurred during last deglaciation, which is preceded by H1 and
derailed by a temporary return to deep freeze in YD (Alley and Clark 1999). Multiple freshwater
fluxes are discerned, which are accompanied by IRD and retain the millennial spacing of DO-
cycles (Keigwin et al. 1991, Fig. 6; Bond et al. 1997, Fig. 6), suggesting their common origin in
the periodic calving of the ice sheet. In addition, there are two massive meltwater pulses (MWP-
1A and 1B) derived from meltback of LIS by the interglacial warmth (Fairbanks 1989).
The meltwater is rerouted from Mississippi to St. Lawrence rivers when LIS has sufficiently re-
treated, which may reinforce the calving-induced freshwater flux to cause YD (Broecker et al.
1988; Teller 1990; Marchitto and Wei, 1995). The coldness of YD however halts MWP-1A as
seen in the glacial readvance (Broecker et al. 1988), resulting in only small overlap between the
two (Lehman and Keigwin 1992). Since LIS has largely disintegrated during the Preboreal,
MWP-1B causes only moderate cooling marking the 8.2 ka event (Alley et al. 1997). As an
added puzzle, the YD-like climate reversal did not occur during penultimate deglaciation (Carl-
son 2008).
As freshening and cooling have opposite effects on marine $\delta^{18}$O, their relative importance would
muddle the interpretation of this data (Keigwin et al. 1991). Both H1 and YD however manifest
strongly in the ice-core $\delta^{18}$O because of the extremely cold winter air (Denton et al. 2005), and
the shutdown of MOC during these events has caused Antarctic warming and rising global $pCO_2$
(Broecker 1998; Shakun et al. 2012), which thus precede the northern climate rebound.
**6.2 Genesis**
We illustrate the deglaciation in the phase-space diagram in Fig. 9 and its time progression in
Fig. 10. Since the freshwater flux is derived from the same source as that of DO-cycle, it is set
as a square wave of 2 ky period (light-shaded), which turns out to produce deglaciation events
like the observed ones hence as labelled. We have drawn two meltwater pulses caused by the in-
terglacial warmth based on observation and the timing constraint noted above. Since the summer
insolation has risen from the deep glacial, we set $q = .8$ (that is, annul absorbed solar flux is 80
$Wm^{-2}$ below the global mean) so the temperature (deficit) and density (surplus) curves are low-
ered from those of Fig. 5 and for simplicity we set the same freshwater flux of .1 Sv.





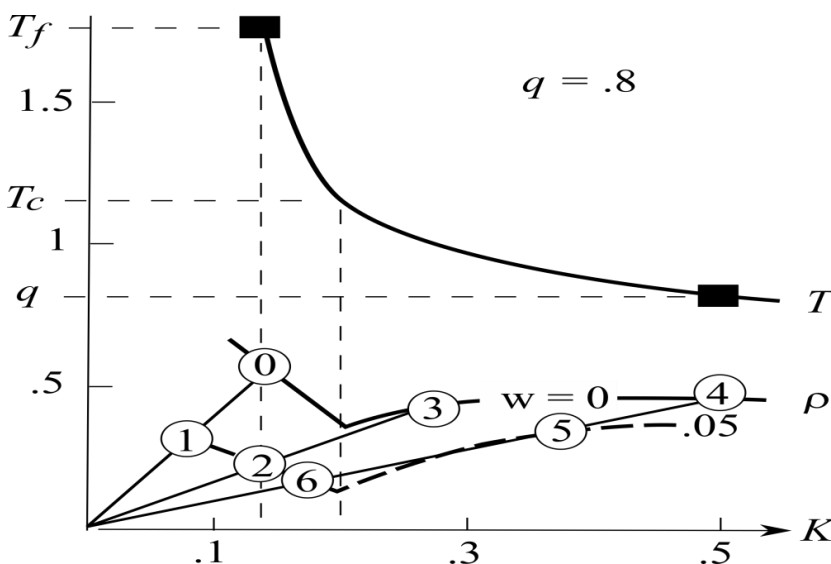

**Figure 9**: Same as Fig. 5 but for $q = .8$, depicting deglaciation sequence.

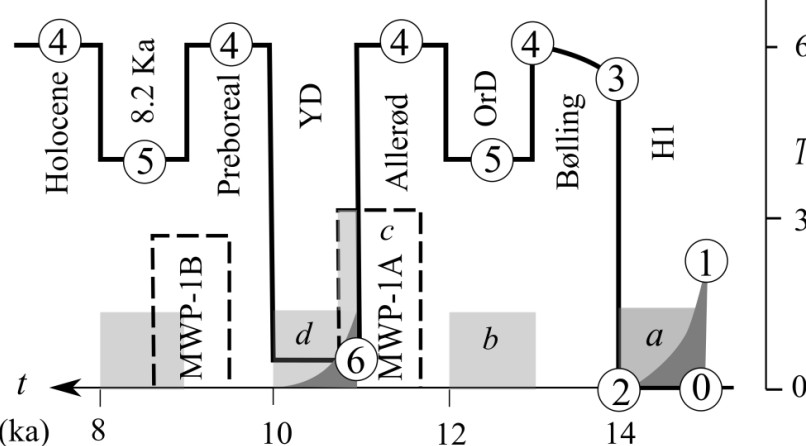

**Figure 10:** Time progression of the deglaciation shown in Fig. 9 with corresponding observed events labelled. Light shades are freshwater fluxes of a square wave of 2-ky period with letters *a-d* corresponding to meltwater events identified in Keigwin et al. (1991). Dark shades indicate sea-ice covers.





The climate signal begins with the glacial MEP (State 0) and runs through numbered states. The
transition from State 0 to 1 to 2 are like the H-cycle, but because of the rising summer insolation,
the MOC line flattened by HE no longer intersects the density curve in the cold branch, so the
climate would vault into the warm branch (State 3), marking the initial deglaciation. Equating
the post-HE temperature Eq. (15) with the convective bound Eq. (5), we derive the criterion for
the deglaciation
$$\frac{w}{q_e} \geq 1 - \left(\frac{T_c}{T_f}\right)^2 . \tag{17}$$

The rhs depends on global-mean temperature and convective flux, which set the long-term super-
orbital condition; the lhs on the other hand depends on both freshwater perturbation and orbital
forcing: a greater freshwater flux would cause deglaciation even with lower summer insolation.
With the standard parameters listed in Appendix, the rhs is .59; and for meltwater flux of .1 Sv,
the deglaciation would occur when annual absorbed SW flux reaches about 81 $Wm^{-2}$ below the
global mean, consistent with that seen in Fig. 9. In comparison, the orbital forcing needs to be 8
$Wm^{-2}$ higher without HE; but since such insolation increase is nonetheless attained in about a
millennium, HE may not delay (McCabe and Clark 1998) --- and possibly even hasten --- the de-
glaciation. On the other hand, since recurring time of HE is shorter than the half precession cy-
cle (10 ky), it always punctuates the deglaciation, as is the observed case (McManus et at. 1999,
Fig. 4).
Following the initial deglaciation, State 3 would propel to State 4 (the interglacial MEP), a tran-
sition spanning the Bølling interstadial. The warmth would elevate the snowline to cause calving
of the marginal ice after a millennium, just like DO-cycle, and the resulting freshwater flux
would cool State 4 to 5 marking Older Dryas (OrD), which however remains an interglacial. The
continuing Allerød warmth (State 4) would melt back LIS to generate massive MWP-1A, but
differing from the northern calving, the meltwater does not perturb the climate until the ice mar-
gin has sufficiently retreated to allow the meltwater to be rerouted to the Hudson Bay. And then
it would reinforce the northern calving by further lowering the density curve (not drawn) to vault
State 4 to 6 marking the YD. Unlike calving paced by internal ice dynamics, the meltback
would be halted by the cold YD, as seen in the glacial readvance. Here for simplicity, we have
neglected the lesser pivot of the MOC line during YD because the small density surplus has in
effect rigidified the MOC line (Ou 2018), so the termination of the millennial YD would rebound
the climate from State 6 to 4, the latter corresponding to the Preboreal. The recurring ice calving
after a millennium causes the 8.2 ka cooling event (State 5), which is analogous to OrD, and
since LIS has largely disintegrated during Preboreal, MWP-1B is insufficient to cause the glacial
flip. After the 8.2 ka event, the climate returns to State 4 corresponding to the Holocene.
**6.3 Synthesis**





The ultimate driver of deglaciation is the rising summer insolation during increasing eccentricity
(Ou 2022b), the suborbital deglaciation events however can be explained by the interplay of
three distinct sources of the freshwater perturbation. The first is calving of the inland ice that
triggers H1, whose post-event warming would vault the glacial into interglacial. The latter sets
up an ablation zone to enable the second source: a periodic calving of the marginal ice, just like
that drives the millennial DO-cycle, whose stades can be identified with OrD, YD and 8.2 ka
event. The glacial flip of YD however requires a third source: the rerouting of the meltwater
generated by meltback of LIS (MWP-1A). Since rerouting occurs only after LIS has sufficiently
retreated, and the cold YD would halt the meltback, there can only be small overlap between YD
and MWP-1A (Duplessy et al. 1992; Lehman and Keigwin 1992). This timing mismatch has
raised question about their causal linkage (Fairbanks 1989), which however is resolved here by
the combined effect of second and third freshwater sources. Since YD involves happenstance of
rerouting and remnant of meltwater, such dramatic climate reversal is not inevitable and indeed
did not occur during MWP-1B or penultimate deglaciation.
As YD is accompanied by strong freshening and cooling, they could cancel each other to leave
little imprint on the marine $\delta^{18}$O data, H1 however is initiated from a glacial state, so freshening
dominates to register in this data (Duplessy et al. 1992, Fig. 1). Both YD and H1 however mani-
fest strongly in the ice-core $\delta^{18}$O because of the extremely cold winter air insulated from ocean
heat by extensive sea ice (Denton et al. 2005). The first three freshwater fluxes and MWP-1A
shown in Fig. 10 can be identified with the four meltwater events discerned in Keigwin et al.
(1991, their Fig. 6), and our model offers a plausible interpretation of the diverse marine $\delta^{18}$O
signal: events $a$ and $d$ are associated with strong cooling (as symbolized by the shaded sea-ice
cover) to cause maxima in the data, but events $b$ and $c$ involve little cooling hence dominated by
freshening to yield minima.
While YD is triggered by freshwater flux, its freshening is due primarily to the MOC shutdown
(Duplessy et al. 1992), which would sequester the southern heat to cause Antarctic warming
(Broecker 1998; Stocker 2000). As such, the latter and the accompanying rising pCO2 precede
the northern climate rebound (Shakun et al. 2012), which however are not causal since the north-
ern deglaciation is already underway and only temporarily reversed by YD on account of the in-
ternal ice dynamics. The termination of YD is accompanied by doubling of accumulation, which
has been attributed to atmospheric circulation change (Alley et al. 1993) but it may simply reflect
the more moist interglacial air, just like that induced by global warming.
That the enhanced moisture transport by global warming may shut down MOC, like the trigger-
ing of YD, is a topic widely examined in the past (Manabe and Stouffer 1999; Rahmstorf et al.
2005; Ou 2018). Model intercomparisons however show considerable uncertainty in the fresh-
water threshold, which can be assessed from our model. As the northern summer insolation has
dimmed since about 10 ka (Alley and Clark 1999), it would raise the freshwater threshold based
on our phase-space diagram (comparing Figs. 5 and 9), but even the lower threshold and the



massive MWP-1B at 8 ka have caused only moderate cooling, we envisage therefore little pro-
spect of a glacial flip; the next glaciation is likely gradual, evolving over millennial timescale
(Ou 2022b), just like previous ones.
**7. Conclusions**
We integrate our ice-sheet and climate models to address abrupt climate changes pertaining to H-
and DO-cycles as well as the last deglaciation punctuated by YD. Since they are all accompa-
nied by IRD, we posit a common source of freshwater perturbation from periodic calving of the
ice sheet due to thermal switch at its bed. We however distinguish two different sources of the
thermal switches: one from trapping of the geothermal heat, which would calve inland ice that
causes HE; the other from surface melt of the ablation zone, which would calve the marginal ice
to drive the DO-cycle. Since ablation zone can be set up in the glacial time only by post-HE
warmth, the glacial DO-cycle is preconditioned on HE to form the Bond cycle; in contrast, the
ablation is already operative during interglacial, so there is self-sustaining Holocene DO-cycle
even in the absence of large ice sheet or HE. In addition to the thermal switch paced by internal
ice dynamics, the meltback of LIS by interglacial warmth would generate massive meltwater
pulses and if the meltwater is rerouted to the Hudson Bay, it may augment calving-induced fresh-
water flux to cause YD, a dramatic reversal of the deglaciation.
Since the thermal switch operates on extremely short (years) subglacial hydrological timescale,
the resulting freshwater flux is step-like, and then ocean responds to this flux on decadal time-
scale, they together would endow abruptness to these climate signals, which thus need not in-
volve ocean mode change, as commonly assumed. Since recuring calving is constrained by mass
balance, it is naturally availed millennial timescale. This timescale however is not inherent to
the ocean and may emerge in numerical simulations only under tenuous external condition or
stimulus, which is at odds with robust time signature of the observed signals.
In addition to the common origin in the ice calving depicted in our ice-sheet model (Ou 2022a),
our climate model (Ou 2018) has underscored a key process in the ocean response. Recognizing
that MOC is subjected to random eddy shedding across the subtropical front, we apply probabil-
ity law of the NT to deduce that it would be propelled on millennial timescale toward MEP. As a
direct consequence of this MEP adjustment, termination of HE is characterized by a sudden
warming to be followed by gradual cooling to exhibit saw-toothed H-cycle, and then the cooling
trend would anchor DO-cycles to form the hierarchical Bond cycle. The YD, just like other
freshwater perturbation, is paced by the internal ice dynamics but it requires reinforcement from
the rerouted meltwater to cause the glacial flip. As it involves happenstance of rerouting, such
dramatic climate reversal did not occur during the penultimate deglaciation.
By incorporating the calving origin of the freshwater perturbation and the MEP adjustment of the
ocean, our theory has provided an integral account of abrupt climate changes. There is no need
to invoke disparate or extraneous physics in explaining their diverse features, as seen in our syn-



thesis of previous works. The robustness of deduced features and their resemblance of the ob-
served ones suggest that our theory might have isolated the governing physics of abrupt climate
changes.
**Appendix**
$a_s$    Aspect ratio of surging ice
$\dot{a}$    Accumulation ($= .1 \, m \, a^{-1}$)
$C_{p,o}$    Specific heat of ocean ($= 4.2 \times 10^3 \, J \, Kg^{-1} \, {}^0C^{-1}$)
$g$    Gravitation acceleration ($= 9.8 \, m \, s^{-2}$)
$\dot{g}$    Geothermal flux ($= 6 \times 10^{-2} \, Wm^{-2}$)
$h_s$    Ice height at surge termination
$[h]$    ELA for DO-cycle ($= .5 \, km$)
$K$    MOC mass flux
$[K]$    Scale of $K$ ($= \alpha^* l (2\rho_o C_{p,o})^{-1} = 6 \, m^2 s^{-1}$)
$l$    Latitudinal span of subpolar ocean ($= 4 \times 10^3 \, km$)
$L$    North Atlantic basin width ($= 6 \times 10^3 \, km$)
$q$    Cold-box deficit of absorbed solar flux
$q_e$    Excess forcing over warm-transition threshold
$[q]$    Scale of $q$ ($= 100 \, Wm^{-2}$)
$\bar{q}_c$    Global convective flux
$S$    Cold-box salinity deficit
$S_0$    Reference salinity ($=35$)
$[S]$    Scale of S ($= \alpha[T]/\beta = 1$)
$t_{ratio}$    ratio of surge/creep duration
$[t]$    Timescale for DO-cycle ($\equiv [h]/\dot{a} = 5$ ky)
$T$    Cold-box SST deficit
$T_c$    Convective-bound temperature
$T_f$    Freezing-point temperature
$[T]$    Scale of $T$ ($= [q]/\alpha^* = 8 \, {}^0$C)
$\bar{T}$    Global-mean SST ($=14 \, {}^0$C)
$\Delta T$    Temperature range of H-cycle
$w$    Freshwater flux
$[w]$    Scale of $w$ ($= 2[K][S]/S_0 = .34 \, m^2 s^{-1}$)
$\alpha$    Thermal expansion coefficient ($= 10^{-4} \, {}^0C^{-1}$)
$\alpha_h$    Heating parameter ($=.48$)
$\alpha^*$    Air-sea transfer coefficient ($= 12.5 \, Wm^{-2} \, {}^0C^{-1}$, Ou 2018)
$\beta$    Saline contraction coefficient ($= 8 \times 10^{-4}$)
$\rho$    Cold-box density surplus
$[\rho]$    Scale of $\rho$ ($= \rho_o \alpha[T] = .8 \, Kg \, m^{-3}$)
$\rho_i$    Ice density ($= .92 \times 10^3 \, Kg \, m^{-3}$)
$\rho_o$    Reference ocean density ($= 10^3 \, Kg \, m^{-3}$)



$\mu$        Moisture parameter (= 0.3)

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
