# Peer review of "A theory of abrupt climate changes: their genesis and anatomy"

_EGUsphere, 2022_

## Author Comment (AC1)

EGUsphere, referee comment RC1
https://doi.org/10.5194/egusphere-2022-208-RC1, 2022

[Figure]

**Comment on egusphere-2022-208**

Anonymous Referee #1
* * *
Referee comment on "A theory of abrupt climate changes: their genesis and anatomy" by
Hsien-Wang Ou, EGUsphere, https://doi.org/10.5194/egusphere-2022-208-RC1, 2022
* * *
The manuscript by Ou presents a theory of abrupt glacial climate changes. The author
postulates that Heinrich and Dansgaard-Oeschger events have the same nature and are
related to abrupt changes in the ice calving into the ocean. In fact, the idea that abrupt
climate changes of the glacial age were related to strongly variable freshwater flux into
the ocean is by far not new, although one of many and not the most popular now. The
only novelty of the proposed theory is that, according to the author, abrupt climate
changes do "*not involve ocean mode change, as commonly assumed*"  (L. 11). Although
the author repeated this statement a dozen times, he did not explain what he
understands under "mode change", and this is why it is difficult to assess how much the
mechanisms proposed by the author differ from those that have been proposed in
numerous previous studies.

1)  *Thank you for the highly substantive comments, my responses are given below in
    italics.*

2)  *Novel physics of the theory includes distinguishing two thermal switches and a
    ocean closure based on maximum entropy production, both justifiable by
    observations and numerical calculations.  They have led to a single dynamical
    framework integrating all three abrupt changes, which moreover has reproduced
    their salient features as exemplified in Figs. 6, 8 and 10.*

3)  *The only distinctive ocean modes (not subtler variants) contained in numerical
    models are the bistable IG/G first discovered by Manabe and Stouffer (1988),
    which are characterized by on/off MOC.  I will add this specification in Introduction.*

The main problem of the manuscript under consideration is that it completely ignores a
vast amount of recent studies on abrupt climate change. Excluding self-citations, only two
(!) cited papers were published during the past ten years and the absolute majority of
cited papers were published in the past century. This "statistic" is in odd with the drastic
increase in the number of empirical and modelling studies of abrupt climate changes in
recent times. This ignorance about the contemporary progress in understanding the
mechanisms of past climate change results in the numerous erroneous statements and
premises on which the theory is built, which made the manuscript under consideration
totally inappropriate for publication in scientific journals.

4)  *Since the paper addresses three phenomena (H, DO and YD), I chose to include
    only primary references, which are necessarily biased toward original (hence
    earlier) contributions.  Based on your comments, I will boost references in
    Introduction (see also response 5).*

**General comments**

The lack of up-to-date knowledge about the progress in the understanding of abrupt climate change gravely affects the entire manuscript. Normally, the ==introduction== presents the current status of the progress in the related subject and the motivation for the presented study. Instead, the author used the introduction to introduce his own theory of abrupt climate changes. Already the first paragraph contains numerous inaccurate and erroneous statements:

> 5) *Somewhat unconventional ordering is related to the scope of the paper dealing with three phenomena, so I figure that discussions of previous works would be more effective (and less repetitive) if they are presented in the synthesis section of the individual topics where they can be contrasted with our model deductions. I however agree with your comment and will boost Introduction to sharpen the objective of the paper.*

The author claims that both Heinrich events and Dansgaard-Oeschger cycles *"are all accompanied by ice-rafted debris (IRD) ... suggesting a common origin in the calving of ice sheet due to thermal switch at its bed"*. The author referred here to a ==very old paper== (Bond et al. 1997), based on a very coarse resolution record. In fact, the situation with IRD during Heinrich and non-Heinrich stadials is very complex. The late are found only in some locations and may imply better survival of icebergs in colder climates rather than an increase in their production. Moreover, there is a potential ==time lag== of IRD compared to the onset of stadials (e.g. Barker et al., 2015), which contradicts the idea of freshwater forcing of stadial events. Thus, it is very likely that IRD during stadials represents the response rather than the forcing of abrupt climate changes.

> 6) *Bond et al. (1997, their Fig. 6) show that glacial and Holocene DO have similar IRD signal (in both amplitude and period), which transcends age resolution. This commonality supports our postulated calving of the marginal ice in the ablation zone, which would be operative even during glacial by the post-H warmth. I should stress this point in revision.*

> 7) *In addition to Bond et al. (1997), more recent studies (Elliot et al. 2002, Fig. 3, perhaps the best reference) show strong correlation between IRD and SST in DO while linkage of MOC and SST cannot be discerned (in contrast to the H-cycle). This observation supports ice calving as the origin of the DO, and not the other way around (a case also made strongly by Van Kreveld et al. 2000).*

> 8) *Although there could be a lag at the northern site of Barker et al. (2015), there are many possible explanations without negating the foregoing causality: the cold stadial would delay the ice melt; the ice melt should precede dislodging of IRD; icebergs could be jammed in the Denmark Strait; icebergs reach the northern site later than the southern site (no lag there) following the Irminger Current. Barker et al. (2015) offered instead circuitous explanations, such as slow/fast cooling and cooling-induced calving, which are less convincing.*

The next sentence (L. 35) - *"since recurring time of calving is constrained by ice mass balance, the resulting freshwater flux is naturally availed the millennial timescale, a timescale not inherent to the ocean"* - is even more problematic. First, the author does not explain why ==recurring time== of calving is constrained by ice mass balance and why it should be millennia. The last part of the same sentence, namely, that the millennial time scale is ==not inherent== to the ocean, is just wrong. Numerous recent studies demonstrate that the ocean alone can produce millennial-scale self-sustained oscillations resembling Dansgaard-Oeschger events (Peltier and Vettoretti, 2014; Brown and Galbraith, 2016; Klockmann et al., 2018). In fact, this finding is not so new: the existence of millennialscale self-sustained AMOC oscillations has been demonstrated already in Winton and Sarachik (1993).

> 9) *I have provided a detailed discussion of timescale in my 2022 paper on glacier instability (Section 3.5, attached). Basically, it is the ice thickness (set by geothermal heat or ELA) divided by accumulation. I agree with your comment and will add a discussion of this timescale in connection with Fig. 1.*

> 10) *The timescale of ocean mode-change depends on the strength of the kick by either hosing or unbalanced initial state, the latter includes the three papers you*

*mentioned. And for damped oscillation, the timescale is the ocean overturning time, which can be centennial if MOC is weak, as seen in Winton and Sarachik (1993, also discussed in my 2012 AMV paper, section 1). To the degree that it depends on the forcing timescale, it is "not inherent" to the ocean --- in contrast to the internal ice-sheet clock, but to avoid misunderstanding, I will remove or rephrase this statement.*

The first paragraph ends with another erroneous statement: *"variation in the sea-surface temperature (SST) remains well short of mode change except during deglaciation"*. The authors did not explain which model is meant here, but I guess this is the "ocean mode." Then, the author explains what "short" means: *"SST variation is considerably smaller, ranging in low single digits"* (L. 228). In fact, SST variations in the Northern Atlantic during DO events were about 5°C (e.g. Martrat, 2007; Alonso-Garcia et al., 2011) which is fully consistent with what climate models simulate in response to the ocean mode change, namely a complete AMOC shutdown (e.g. Jakson et al., 2015)

> *11) Martrat (2007) examines the centennial oscillation at mid-latitudes and Alonso-Garcia et al. (2011) consider the precession-induced G/IG cycles in mid-Pleistocene; both are not the millennial DO. Jackson et al. (2015) shut down the MOC to examine its effect on the European climate, which is not relevant to our study.*

> *12) Elliot et al. 2002 (Fig. 3) show that SST signal of the DO is encased within H-cycle, and the latter remains a glacial phenomenon; that is, post-H warmth remains short of IG, as seen also clearly in Bard (2002, Fig. 2). I shall replace the "low-single digit" phrase with above statement.*

> *13) The H-related MOC change is substantial, but even post-H MOC remains below the IG strength, and then DO is not associated with discernable MOC signal (Elliot et al. 2002, Fig. 3); both sharply at odds with on/off MOC produced in numerical models.*

In the next paragraph, the author postulates that H-cycle is related to *"to calving of the inland ice"*, whilst *"for DO-cycle it is the surface melt over the ablation zone that causes calving of the marginal ice"*. Unfortunately, the author does not explain why he believes thy HE events are associated with calving of "grounded ice" and what "marginal ice" means in the case of DO events.

> *14) Only inland ice near the ice divide as implicated in MacAyeal (1993) may inject large freshwater flux of the H-cycle (amounting to a sea-level change of order 10 m). I thought it is understood that if the thermal switch is sited under the equilibrium line, it would carve the marginal ice of the ablation zone, but based on your comment, I should make it more explicit in the revision.*

As far as a very stimulating McAyeal's concept of binge-purge oscillations is concerned, the author should be aware that it has been advanced over the past decades significantly from 1-dimension to 3-dimensional case and evolved apart from geothermal and frictional heat, also strain heating, basal hydrology, activation/deactivation waves, etc., (Calov, 2002, 2010; Roberts et al., 2016; Feldmann and Levermann, 2017; Schannwell et al., 2022).

> *15) I am aware of these expansions, as discussed in my 2022 paper on glacier instability (Section 1, see attached). As a theory, my aim is to isolate minimal physics to advance understanding, an approach that is diametrically opposite to the simulation models, which try to include all relevant physics to improve the realism --- often at the expense of understanding and falsifiability. One should also not lose sight that all these models have tuned sliding velocity, which directly impacts the amplitude and period of H-cycle, an empiricism that has limited their prognostic utility --- regardless their outward sophistication. This empiricism is removed in Ou (2022) by global momentum balance, so the model output shown in Figs. 1 and 2 are prognostic. I will add some these discussions in revision.*

On pages 11 and 14, the author, at last, mentioned alternative mechanisms of abrupt climate changes. He wrote: *"H-cycle has been modelled as ocean mode change (Paillard*

1995; Ganopolski and Rahmstorf 2001), which is ==unsupported by observation==". Which observations do not support "mode change", the author did not explain. In turn, I am aware of numerous ==paleoclimate records== which support qualitative AMOC changes during Heinrich events and non-Heinrich stadial (e.g. Lippold et al., 2009; Böhm et al., 2015). In the next sentence, based on Equation (16), the author claims that SST change should be "*proportional to the freshwater perturbation*" (L. 313). This, however, contradicts a vast amount of modelling studies which show a strongly nonlinear response of AMOC and SST to freshwater perturbation. They also demonstrate that abrupt climate changes can be caused by a very gradual forcing or even without any external forcing. Moreover, simulated ==self-sustainable oscillations== resembling DO events have typically periodicity of one to several millennia. Since I have no reason to believe in ==Equation (16)== more than in the results of realistic climate models, I cannot consider the proposed theory as a valid alternative to the modern concepts of abrupt climate changes.

16) *Lippold et al. (2009) cautions that his proxy is insufficient to discern MOC, which undercuts Bohm et al.'s (2015) interpretation of vigorous MOC through glacial cycles.*

17) *Elliot et al. (2002, Fig. 3) show that H-cycle is a glacial phenomenon: both its SST and MOC remain below IG values, contrary to its numerical simulation anchored on G/IG.  For DO, its SST is encased within H-cycle to form Bond cycle, and its MOC signal is undetectable.  The DO period in numerical simulations is tuned (see response 10), which moreover have not reproduced the characteristic Bond cycles. Given these deficiencies of the numerical models, they may not serve as an arbiter of our theory, and both should be tested against observations.  On this score, it worth stressing that our deduced signals shown in Figs. 6, 8 and 10 have reproduced salient features of the observed phenomena, a task that remains unfulfilled by numerical calculations.*

18) *To dispel a wide-spread misconception, it should be pointed out that the large SAT signal of DO seen in ice-core data should not mask its muted SST signal (see response 12).  Denton et al. (2005) have convincingly argued that such large SAT signal simply reflects the extremely cold winter air caused by extensive winter sea ice when SST is hovering around the freezing point (Li et al. 2010).*

19) *The statement on Eq. (16) serves only to indicate that our H-cycle is not a mode change, but to avoid misunderstanding, I will rephrase this sentence.*

Three pages later, the authors criticized results obtained with very simplistic models (Sakai and Peltier, 1999; Schulz and Paul, 2002), but similar results have been later obtained with much more realistic climate models (see above). Interestingly, here the author explicitly assumes that there were Dansgaard-Oeschger events during the Holocene, which is, of course, ==wrong== - the last DO event (Bolling-Allerod) occurred well before the onset of the Holocene.

20) *Holocene DO is well documented (Bond 1997; Schulz and Paul 2002), which has similar period as its glacial counterpart.  Bolling-Allerod punctuated by OrD may indeed be interpreted as DO interstadials, so are the warm periods of Holocene.*

As far as the explanation of the deglaciation is concerned, one only can wonder why the author placed meltwater pulses 1A and 1B ca. ==3000 years later== than they happened in reality (Fig. 10).

21) *Fig. 10 is our interpretation of deglaciation sequence based on a square wave of 2 ky period, which obviously is too crude for a tight fit.  We assign MWPs at 11-12ky and 9 ky, similar to observed ones (12 and 9ky, Fairbanks, 1989), there is not a 3 ky lag.  In addition, our sequence matches the four freshwater fluxes of Keigwin et al. (1991, labelled a-d in Fig. 10).*

22) *In conclusion, I appreciated your detailed comments, which have pointed to obvious shortfalls of the paper, and I very much like to know if they have been adequately addressed.  I found the discussion to be highly stimulating, which has sharpened my own thinking, and would welcome additional input from you.*

**References**

Alonso-Garcia, M., et al. Ocean circulation, ice sheet growth and interhemispheric coupling of millennial climate variability during the mid-Pleistocene. Quaternary Science Reviews 30, 3234-3247, 2011.

Barker S., et al. Icebergs not the trigger for North Atlantic cold events. Nature, 520, 333-336, 2015.

Böhm, E., et al. Strong and deep Atlantic meridional overturning circulation during the last glacial cycle. Nature, 517, 73-76, 2015.

Brown, N., and Galbraith, E. D. Hosed vs. unhosed: interruptions of the Atlantic Meridional Overturning Circulation in a global coupled model, with and without freshwater forcing. Climate of the Past, 12, 1663-1679, 2016.

Calov, R., et al. Results from the Ice-Sheet Model Intercomparison Project – Heinrich Event INtercOmparison (ISMIP HEINO), J. Glaciol., 56, 371–383, 2010.

Feldmann, J. and Levermann, A. From cyclic ice streaming to Heinrich-like events: the grow-and-surge instability in the Parallel Ice Sheet Model, The Cryosphere, 11, 1913–1932, 2017.

Jackson, L. C., et al. Global and European climate impacts of a slowdown of the AMOC in a high resolution GCM. Climate dynamics, 45, 3299-3316, 2015.

Klockmann, M., et al. Two AMOC states in response to decreasing greenhouse gas concentrations in the coupled climate model MPI-ESM. Journal of Climate, 31, 7969-7984, 2018.

Lippold, J. et al. Does sedimentary 231Pa/230Th from the Bermuda Rise monitor past Atlantic meridional overturning circulation? Geophys. Res. Lett. 36, L12601, 2009.

Martrat, B., et al. Four climate cycles of recurring deep and surface water destabilizations on the Iberian margin. Science 317, 502-507, 2007.

Peltier, W. R., and Vettoretti, G. Dansgaard-Oeschger oscillations predicted in a comprehensive model of glacial climate: A "kicked" salt oscillator in the Atlantic, Geophys. Res. Lett., 41, 2014.

Roberts, W. H. G., et al. The role of basal hydrology in the surging of the Laurentide Ice Sheet, Clim. Past, 12, 1601–1617, 2016.

Schannwell C., et al. Sensitivity of Heinrich-type ice-sheet surge characteristics to boundary forcing perturbations. Climate of the Past Discussion, 2022.

Winton, M., and Sarachik, E. S. Thermohaline oscillations induced by strong steady salinity forcing of ocean general circulation models. Journal of Physical Oceanography, 23, 1389-1410, 1993.